# Cyclic Behavior of Autoclaved Aerated Concrete External Panel with New Connector

**DOI:** 10.3390/ma15248778

**Published:** 2022-12-08

**Authors:** Jianhua Cui, Shulin He, Kewei Ding, Yu Zhang, Xiaoying Kong

**Affiliations:** 1School of Civil Engineering, Anhui Jianzhu University, Hefei 230601, China; 2School of Civil Engineering, Guangzhou University, Guangzhou 510006, China

**Keywords:** connector, external ALC panel, finite element simulation, parameter analysis, seismic performance

## Abstract

In this paper, a new slip-type crossing connector is proposed for autoclaved aerated concrete (ALC) panels with steel frames, and the proposed connector is also studied deeply in terms of seismic performance. The research included pseudo-static tests and finite element simulations. First, the seismic performance of slip-type crossing connectors and standard L-hooked bolts was studied comparatively, including the stability, bearing capacity, stiffness, energy dissipation, and hysteresis performance. ABAQUS 2020 software was used to establish finite element models, and the results of the experiments were verified with simulations on the basis. According to the simulations, a parameter analysis of connector optimization was carried out. The effects of connector thickness and connector plate length on the seismic performance were further investigated. From the experimental and simulation results, the slip-type crossing connector has excellent performance and good assembly efficiency, it can improve the deficiencies of the existing connectors. The comparison demonstrated that the slip-type crossing connector has a complete hysteresis curve, a high energy dissipation capacity, and a 9.7% increase in bearing capacity. The appropriate reduction in connector thickness and plate length can ensure superior seismic performance while saving resources. The finite analysis method can guide the design and implementation of new external ALC panel connectors.

## 1. Introduction

With the transformation and upgrading of traditional industries and the innovative development of residential systems, prefabricated building systems have become a promising candidate to meet the requirements of green buildings with good seismic performance and repairability while saving more resources [1,2]. Although prefabricated buildings offer numerous economic, environmental, and social benefits, they have received a very low level of adoption in the global construction industry, despite their advantages [3]. In recent years, the assembled steel building system has been widely developed under the promotion of the state. Still, the development of assembled steel residential enclosure systems is lagging behind. Implementing a highly integrated and industrialized enclosure system can be used to address this lagging problem [4]. In the current circumstances, precast concrete elements have provided a solution to the conventional challenges associated with steel structures. Many scholars have concentrated on the seismic performance of various precast concrete elements and have demonstrated their various mechanical qualities [5,6,7,8]. The cross-application of different panels has developed a new type of enclosure system [9]. Nowadays, autoclaved aerated lightweight concrete (ALC) panels are commonly used in precast steel structures due to their superior physical properties as internal and external partitions of the enclosure system [10]. Compressive properties of ALC panels have been reported [11,12]. Many experts have examined the other features of ALC panels, showing that autoclaved aerated concrete exhibits good functional performance and dynamic characteristics [13,14,15]. Although ALC panels are usually considered non-structural components, interaction performance still exists between these panels and boundary steel frames under seismic loads. Increasing evidence suggests that most assembled enclosure panels produce large deformation and fall off under destructive earthquakes [16]. Therefore, investigation of the dynamic reaction of enclosure panels’ attachment to steel frames during an earthquake is required. In prefabricated high-rise steel structure buildings, the connection behavior and structural measures of the walls and main structure are related to the safety and stability of the whole structure [17]. The development of new connectors should ensure good deformation coordination ability and seismic performance of the wall panels and the main structure during earthquakes.

On this basis, the study of the connection method of ALC panels to the main structure has become a top priority. At present, studies on the seismic performance of external panel connectors are still limited. In the research of assembled steel frame structures, the standard rigid connector between an ALC panel and a steel frame is mainly an L-hook bolt [18]. Although an L-hook bolt can allow the panel to rotate during an earthquake, it cannot coordinate the relative displacement between the panel and the steel frame well. Cao et al. [19] developed a flexible connector consisting of welded studs, angles, and embedded parts, which can be used as both a lower and upper connector for external panels. Although experimental results have shown that embedded connectors have good force performance and application prospects, their design is complicated and unfavorable for installation. To solve the related problem, Ding et al. [20] developed a pendulous ALC Z-panel connector, and the test showed that the new connector could better realize the swinging function of the panel under the reciprocating load and could reduce the damage of panels, while the manufacturing process of the connector is complicated. Zhang et al. [21] developed a steel pipe anchor slip connector compared with a conventional steel anchor connector by simulated seismic shaking table tests of steel frames with a five-story 1/4-scale. The results demonstrated that the connector could effectively protect the panels through reserved long holes and better synergize with the main structure during earthquakes by relying on the slip mechanism of the connector. However, the precast component of the steel pipe anchor connector is vulnerable in the process of transportation and requires high construction accuracy.

In order to solve the relevant deficiencies of the existing connectors, this paper proposes a new flexible connector—a slip-type crossing connector, which can solve the existing deficiencies well while having better seismic performance. The connector has low installation difficulty, and the limitation hole was also designed as a long round hole, allowing the bolt to carry out limited movement in large displacement. Relative displacement may exist between the external panel and the main structure [22]. No large internal force is generated under external load that would lead to connector damage, which can reduce the phenomenon of panel crack generation or the risk of panels falling off. To verify the reliability of this new connector, seismic performance-related curves were obtained through two sets of low cyclic reversed loading test results and related numerical simulation comparison analysis [23]. Additionally, we used the numerical simulation software ABAQUS to systematically analyze the slip-type crossing connector under varying thicknesses and plate lengths. According to the investigation, the slip-type crossing connector is fully functional and reliable and can replace the existing L-hooked bolt in high-rise steel constructions.

## 2. Materials and Methods

### 2.1. Properties of Materials

Table 1 presents the material properties. According to GB/T228.1-2010 ’Metallic Materials-Tensile testing’ [24], the yield stress (*ƒ_y_*), the ultimate stress (*ƒ_u_*), and the elongation at fracture (*δ*) of the tensile coupons cut from steel sheets (used in the beams, columns, and connector) were measured, as illustrated in Figure 1a. According to GB/T 11959-2020 [25], the ALC’s average compressive strength and modulus of elasticity through testing were 3.56 and 1.77 N/mm^2^, respectively, given in Figure 1b.

### 2.2. Experimental Design

The test frame was a single-story, single-span, flat combination frame, as shown in Figure 2a. The beam was made of an H-beam with a cross-sectional size of 240 × 175 × 7 × 11 mm, while the column was made of an H-beam with a cross-sectional dimension of 200 × 200 × 8 × 12 mm. The steel structure material was Q235 grade hot-rolled steel. The bolts at the beam-column joint were 10.9-grade M24 high-strength bolts. The thickness of the ALC panel, the panel connection method, and the connection part are shown in Table 2. The detail of the slip-type crossing connector is illustrated in Figure 2b.

The ALC panels’ connection type included an L-hooked bolt and a slip-type crossing connector. The detailed connection measurements of external ALC panels to assembled steel beams are shown in Figure 3a,b. Figure 3c,d shows the site of testing.

### 2.3. Test Setup and Loading System

Figure 4 shows the test loading devices, and the test was conducted at Anhui Province’s Key Laboratory of Building Structure and Underground Engineering. The horizontal load was applied to the structural frame by MTS, based on the current Chinese code JGJ/T 101-2015 ‘Specification for the seismic test of structures’ for the loading history of all specimens [26]. Figure 5 presents the test loading system. The control of displacement was employed for loading. Three cycles were enforced at each horizontal displacement level of 5–30 mm, and two cycles at each displacement level of 40–120 mm.

## 3. Results and Discussion

### 3.1. Test Phenomenon

#### 3.1.1. Specimen JD-1

The tested specimen, JD-1, was a steel frame with external ALC panels, and L-hooked bolts were used to attach the panels to the frame. Small cracks appeared in the jointing mortar between the first and second panels when the displacement reached 6 mm. Vertical cracks appeared and expanded in the jointing mortar between the panels as the displacement arrived at 8.6 mm, and the destruction of the jointing mortar was accompanied by the dislocation of the panels at 15 mm (shown in Figure 6a). Under the displacement of 40 mm, a weld crack appeared between the steel angle and the L-hooked bolt at the upper end of the second panel, and the L-hooked bolt had a rotation relative to the steel angle at the first cycle of 75 mm (seen in Figure 6b). When the displacement reached the second cycle of the 75 mm stage, the panel angle ruptured due to extrusion with the support plate (shown in Figure 6c). A figure-of-eight crack was first produced at the hole of the L-hooked bolt in the bottom of the third panel and the crack width developed by approximately 1.02 mm at the second cycle of 90 mm (illustrated in Figure 6d). Meanwhile, the right upper beam-column welds fractured (illustrated in Figure 6e). When the displacement reached 120 mm, a large area of spalling occurred on the corner of the panel after extrusion (illustrated in Figure 6f).

#### 3.1.2. Specimen JD-2

The tested specimen JD-1 was a steel frame with external ALC panels, and a slip-type crossing connector is used to attach panels to the frame. There was no obvious phenomenon in the beginning stage of specimen JD-2. When the loading reached 30 mm, the upper crossing connector bolt and the long circular hole produced relative sliding, and the bolt moved to the edge of the hole at 40 mm (shown in Figure 7a). Meanwhile, a small amount of powder fell between the wall panels due to extrusion, and the first and second panels started to separate due to in-plane horizontal force, while the gap between the two panels eventually expanded to 30 mm at the stage of 60 mm (shown in Figure 7b). When the displacement reached the 75 mm stage, a crack appeared at the upper bolt hole of the second panel (illustrated in Figure 7c), and the dislocation phenomenon appeared between the wall panel and the bottom support plate surface because of the wall panels’ relative vertical displacement (illustrated in Figure 7d). Under the displacement of 90 mm, the upper end of the fifth panel was crushed and broken at the connector (illustrated in Figure 7e). When the displacement reached the 120 mm stage, all of the jointing mortar between the wall panels was damaged and fell off under large displacement, and the right upper beam-column welds finally fractured (illustrated in Figure 7f).

### 3.2. Test Results and Analysis

#### 3.2.1. Hysteresis Curve

The MTS loading system recorded the hysteresis curve of the horizontal load–displacement relationship at the top of the specimen during the test. Figure 8 shows the hysteresis curve of JD1 and JD2. The hysteresis curves of both specimens were relatively full. The bearing capacity of specimen JD-2 after 60 mm was greater than that of specimen JD-1 under the same lateral displacement conditions. Meanwhile, the overall area of the hysteresis curve enclosure in JD-2 was greater than that in JD-1.

The lateral stiffness of the specimens gradually degraded as the loading displacement increased, and the flat combination frame experienced the process from complete elasticity, through elastoplasticity, to plasticity. During the process, the failure of the jointing mortar reduced the integrity of the external wall panels and weakened the synergy between the overall wall panels and the frame to some extent. Specimen JD-1, which was rigidly connected with an L-hooked bolt, the adhesive mortar between the ALC panels had completely failed under the displacement of 60 mm. In contrast, specimen JD-2 released the harmful displacement early to protect the integrity of the wall panels due to the slip mechanism of the upper connectors and the synergistic action of the external wall panels with the main structure. In addition, when the loading reached the 90 mm stage, the upper beam-column welds of JD-1 fractured, resulting in an incomplete structure. This phenomenon predated JD-2 and resulted in a considerable decrease in the lateral displacement stiffness of the plane frame. Therefore, specimen JD-2 had a higher overall bearing capacity than specimen JD-1.

#### 3.2.2. Skeleton Curve

Figure 9 depicts skeleton curves of the JD-1 and JD-2 specimens, which were generated by connecting the peak points of each loading in the same direction on the hysteresis curve diagram. By comparing the overall trend of the skeleton curves, it can be seen that the trend of specimens’ skeleton curves at the start of loading was essentially identical. Both specimen curves grew by linearity at the beginning stage, and the elastic stiffness was 2.659 and 2.664 kN/mm for JD1 and JD-2, respectively, indicating that the wall panel had good synergy with the main structure and worked together in a coordinated manner. Due to the formation of cracks in the jointing mortar of the wall panels, the slope of the skeleton curve began to drop as the loading displacement increased, and it entered the cracks’ working stage. In the later period, the connection strength between the external wall panel and the main structure decreased, the stiffness degraded continuously, and the specimens’ bearing capacity decreased. Due to the failure of the connector and panel damage, the bearing capacity of specimen JD-1 decreased more obviously in the 90–120 mm stage.

From the characteristic values of the skeleton curves in Table 3, it can be seen that the bearing capacity of specimen JD-2 increased by 9.7% and the ultimate load increased by 5.3%, while the yield force bearing capability of specimen JD-2 noticeably increased by 13.8% and the yield point delayed by 13.6%, respectively, in comparison to specimen JD-1. It indicated that the slip mechanism of specimen JD-2 delayed the cracks from occurrence while ensuring the integrity of the wall panels.

#### 3.2.3. Stiffness Degradation

The specimen under cyclic reciprocal loading, due to the progression of the material’s plastic deformation and the development of cracks in the wall panel, resulting in the same peak load under the peak point as the number of cycles increases, is called stiffness degradation. The secant stiffness was chosen to depict the stiffness degradation of the specimen. It can be calculated according to the following equation [27]:(1)$$K_{i}=\frac{\left(\left|+F_{i}\right|+\left|-F_{i}\right|\right)}{\left(\left|+X_{i}\right|+\left|-X_{i}\right|\right)}$$
where *F_i_* and *X_i_* denote the maximum load and displacement in the i-th loading regime, respectively.

The stiffness degradation curves of specimens JD-1 and JD-2 are shown in Figure 10. It can be seen from Figure 9 that the initial secant stiffness of specimen JD-2 (3.10 kN/mm) was 78.6% of that of specimen JD-1 (3.95 kN/mm). This is because the L-hooked bolts, as rigid connection connectors, limited the relative displacement of the wall panel and the main structure in the initial stage. Thus, the wall panel of specimen JD-1 contributed more to the pre-stiffness. In contrast, the slip-type crossing connector bolt slipped in its limit hole and released the harmful displacement early, while the initial stiffness was less than that of specimen JD-1.

The secant stiffness of both specimens decreased as displacement increased, caused by the development of cracks and plasticity of the material. However, the rate of stiffness degradation was faster in the 0–75 mm stage for JD-1 compared to JD-2. The secant stiffness of JD-2 was already significantly higher than that of JD-1 at 60 mm and the stiffness of JD-1 increased by 10.4% compared to JD-2 at the elastic stage (68.85 mm). This is because the slip mechanism of the new connector made full use of the wall panels’ shear stiffness while the bolts were close to the edge of the limitation hole at the 40 mm stage, which can delay the appearance of panel cracks and slow down the rate of stiffness degradation to some extent. This indicates that the slip-type crossing connector was more likely to maintain the integrity of the frame and wall panel, which had better synergistic working performance.

#### 3.2.4. Energy Dissipation

When a structure is subjected to seismic loading, whether the specimen can absorb the seismic release energy well is an important indicator of its seismic performance [28]. The better the structure’s overturning resistance, the more energy it dissipates. In this paper, the energy dissipation capability was determined by calculating the equivalent viscous damping ratio of the specimens, as shown in the following equation [29] and Figure 10:(2)$$\xi_{e}=\frac{1}{2\pi }\cdot \frac{S_{ABC}+S_{CDA}}{S_{OBE}+S_{ODF}}$$
where *S_ABC_* and *S_CDA_* in the numerator are the areas beneath each curve *ABC* and *CDA*, and *S_OBE_* and *S_ODF_* in the denominator are areas within Δ*_OBE_* and Δ*_ODF_*, as shown in Figure 11, respectively.

As shown in Figure 12, the equivalent viscous damp ratio (*ξ_e_*) of JD-1 and JD-2 was between 0.0175–0.1031 and 0.0322–0.1056, respectively. The trend of curves at the start of loading was linear growth. The damage phenomenon of the ALC external wall panels of specimen JD-1 was more advanced in the instance of larger deformation of the steel frames, so the equivalent viscous damping ratio decreased in the early 0–10 mm of loading. The changing trend of both specimens gradually increased during the loading process; however, *ξ_e_* of JD-2 was larger than that of JD-1 at each level load. Specimen JD-2 dissipated energy mainly through the relative slip between the connectors and the wall panels, while specimen JD-1 only allowed the rotation at the connectors.

The equivalent viscous damp ratio of the new connector increased by 25.5% at the yield stage. The *ξ_e_* of the two specimens tended toward the same at the ultimate state because the wall panels gradually withdrew from work as the displacement increased and the proportion of energy dissipation assumed by the steel frame increased. The equivalent viscous damp ratio of specimen JD-2 was greater than that of specimen JD-1, indicating that the slip-type crossing connector had a higher energy dissipation capacity than the L-hooked bolt.

### 3.3. Finite Element Analysis

The time and cost of full tests can be effectively saved by using finite element models to analyze the proposed connector [30]. In order to accurately reflect the force mechanism of the ALC panel flat combination frames, a series of verifications were performed on the traditional connector (L-hooked bolt) and the new connector, named FEM1 and FEM2.

#### 3.3.1. Finite Element Analysis Model

Based on the proposed external ALC panel steel frames with different connection methods, finite element analysis was performed by ABAQUS software. A hexahedral linear reduction integral solid element (C3D8R) was employed to simulate the ALC panels, connectors, bolts, steel beams, and steel columns [31]. A linear truss element (T3D2) was used to simulate the internal reinforcing bars of the ALC panels. Considering the calculation accuracy of the model, the mesh was encrypted at the holes of the connectors, bolts, and ALC panel holes. The finite element models of two types of external ALC panel steel frames are shown in Figure 13. A tie constraint condition was used to define the constraint relationship between the connectors and the steel beam. The constrained type of the other interaction among flat combination frame, L-hook bolt connector, new connector, steel bolt, and ALC panel was set to hard contact. The friction behavior between the ALC panels used a penalty function with a friction coefficient of 0.3 and the friction coefficient of 0.2 is taken between the ALC panel and connector. The internal double-layered reinforcement was embedded into the ALC panel using the Embed command condition [32].

#### 3.3.2. Material Properties

The ALC panel internal reinforcement was HPB300, the slip-type crossing connector of specimen JD-2 used Q345B, and a 5.6 grade M14 bolt was used for the connector with the ALC panel. The L-hooked bolt of specimen JD-1 was made of Q345B, and the bolt size was M12. The beams and columns used hot rolled steel of Q235 grade. An M24 high-strength bolt was used for the beam and column connections. The plastic damage model of concrete was utilized to simulate the damage to the ALC panel, which could well describe the tension and compression behavior of concrete materials under cyclic load. The tensile and compression damage factors were calculated according to the CDP model [33]. Table 4 summarizes the more relevant material properties.

#### 3.3.3. Boundary Conditions and Loads

Figure 14 illustrates the loads and boundary conditions in the finite element model. The control of displacement was adopted for horizontal cyclic loading. The coupling points RP1, RP2, and RP3 were located at the center extension of the frame beam and the bottoms of both steel columns, respectively. The constraints and loads on the coupling points were specified in accordance with the actual boundary conditions, including rotational constraints and displacement constraints (*δ_x_ = δ_y_ = δ_z_ = u_x_ = u_y_ = u_z_ =* 0) at the bottom of the columns, as well as rotational constraints (*δ_x_ = δ_y_ = δ_z_ =* 0) and displacement constraints in the y and z directions (*u_y_ = u_z_* = 0) at the loading point.

#### 3.3.4. Finite Element Results

Figure 15a,b shows a stress cloud diagram of the specimens in the simulation. The stress concentration points in the model were mainly found near the beam-column nodes and bolt holes, consistent with the damage phenomenon of the test specimen. The results were verified by comparing the finite element analysis results with the experimental results. In the hysteresis curve results, the curves’ “pinch” impact was more pronounced in the experimental results than in the simulation results [34]. The initial stiffness was slightly lower than the value obtained from the simulations. All of the above differences, however, were within a reasonable range, and the finite element simulated results agreed with the experimental results.

Figure 15c,d presents the load–displacement hysteresis of the specimens in the test and simulation. In the experimental hysteresis curve results, the positive loading energy consumption was greater than that of the negative loading, while the residual deformation was smaller than that of the negative loading phase. The ABAQUS finite element simulation results were slightly better than the experimental results due to the influence of the steel frame footing and ALC panel reinforcement slippage. Therefore, the results of the established FEMS model match the experimental results and reflect the seismic performance of the structure more accurately, which can be used for further parametric analysis.

### 3.4. Parameter Analysis

The ALC external panels’ slip-type crossing connectors showed good performance in both the test and simulation and were able to provide better energy dissipation capacity and coordinated deformation capacity for the steel frame as a whole. The specimens’ damage and stress concentration phenomena were mainly concentrated at the beam–column connectors and bolt holes in the test and simulation. Meanwhile, the destruction of the wall panel at the connectors was more serious. This indicates that the connectors had a certain degree of influence on the force mechanism of the ALC panel steel frames. To further study the impact of connector thickness and connector plate length on the seismic performance of the structures, five connectors with different thicknesses and three connectors with different plate lengths were designed to provide references for practical engineering applications.

#### 3.4.1. Connector Thickness

Five different thicknesses of 6, 8, 10, 12, and 14 mm were designed, named FEM3, FEM4, FEM2, FEM5, and FEM6.

Figure 16 illustrates the skeleton curves of the above five connectors with different thicknesses. The specimens were at the elastic stage, and the curves overlapped when the loading displacement was small. The curves of the five groups gradually diverged as the loading displacement increased, indicating that the different stiffness contributions of the ALC wall panel and steel frame in the working phase of the connector produced variability. Among these five specimens, the FEM3 group reached the peak load first, and the ultimate bearing capacity decreased by 4.4% in comparison to FEM2 and by 7.7% compared to FEM6. The skeleton curves were all S-shaped, with a descending segment and roughly the same descending trend on each curve. The results show that the influence of slip-type crossing connectors on the structure’s bearing capacity improved as the thickness increased, but the effect was subtle.

Figure 17 illustrates the stiffness degradation curves of the above five connectors with different thicknesses. The overall trend of these curves is consistent. Before the loading displacement of 30 mm, the stiffness degradation curves of the five simulated groups were not significantly different but diverged after the loading displacement rose. The stiffness contribution of the working phase of the connectors of different thicknesses to the ALC wall panels and steel frames did not vary linearly with increasing thickness. It followed from the parameter analysis of FEM2 and FEM3-6 that the initial stiffness of the specimens with the greater thickness were slightly higher, but the rate of stiffness degradation throughout the loading phase was faster.

Figure 18 shows the equivalent viscous damp ratio of the above five connectors with different thicknesses. Each specimen’s equivalent viscous damping coefficient increased continuously as the structure entered the yielding stage. At the later stage of loading, as the number of cycles of low circumferential reciprocal load increased, the difference between the curves of each specimen got larger. Thus, it is further inferred that with the smaller connector thickness of the planar composite frame, it can better absorb the seismic load in the earthquake. In addition, the energy dissipation capacity is worse if the thickness is greater. It indicated that it would be more reasonable to use the FEM3, which is the most economical solution for engineering.

#### 3.4.2. Length of the Connector Plates

Three different plate lengths of 50, 100, and 150 mm were designed, named FEM7, FEM2, and FEM8.

Figure 19 illustrates the skeleton curves of the above three connectors with different plate lengths. Analyzing the three sets of curves, all three groups of specimens were in the elastic stage at the beginning of loading, and the curves coincided. With the increasing loading displacement, there was some difference in the curve, but it was not obvious, and decreasing segments appeared at a later stage. It was initially inferred that the change in nodal plate length did not contribute significantly to the bearing capacity of the external ALC panel flat combination frame, and other seismic properties need to be analyzed.

To further verify the connector optimization, the stiffness degradation of the three groups was compared. The stiffness degradation curves of the above three specimens with varying plate lengths are shown in Figure 20. The stiffness degradation trends remain the same for the three cases of 50, 100, and 150 mm plate lengths, and the difference in secant stiffness values between the groups was small for the same displacement.

Figure 21 shows the equivalent viscous damp ratio for the above three specimens with different plate lengths. The equivalent viscous damp ratio of each group was compared. No significant differences were found in the equivalent viscous damp ratio of the three simulated groups before the loading displacement of 30 mm, and there was a significant increase after 30 mm. The difference was greatest when the loading displacement was 75 mm. This implies that the FEM7 group had a higher energy dissipation capability than the other two groups. The best energy dissipation capacity is achieved when using 50 mm of plate length. It is further inferred that the proper reduction of the plate length can enhance the specimens’ energy dissipation capability and can be employed as an optimization scheme in engineering.

## 4. Conclusions

In this paper, a slip-type crossing connector was proposed to solve the relevant deficiencies of the existing connectors. Low cyclic loading tests were conducted and ABAQUS software was used to establish FEMs models. To further enhance the seismic performance of the new connector, parametric research of the connector was analyzed. Through analysis of the test phenomena, test data, and simulated results, the following conclusions were obtained:(1)The new connector has better connection behavior and can adapt to larger inter-story displacements. The individual L-hooked bolt became loose and failed to connect with the frame beam at the 75 mm stage, while the new connector worked firmly. The integrity of the panel was great before the slip-type crossing connector reached the edge of the hole. It was proven that the new connector can effectively reduce the damage to ALC panels under minor disasters through the sliding mechanism.(2)The new connector has better seismic behavior. Compared to the standard L-hooked bolt, the bearing capacity of the new connector increased by 9.7%, and the ultimate load increased by 5.3%. The equivalent viscous damp ratio of the new connector increased by 25.5% at the yield stage. Although the initial stiffness of the new connector was smaller, the stiffness degradation rate was slower than that of the L-hooked bolt and increased by 10.4% at the elastic stage.(3)The ABAQUS finite element simulated results agreed with the experimental results. In the hysteresis curve results, the curve’s “pinch” impact was more pronounced in the experimental results than in the simulation results, and the initial stiffness was slightly lower than that in the simulated results. All of these differences, however, were within a rational range, and the analysis results of the FEMs had the reference value.(4)The parametric research revealed that appropriate reduction of the plate length and thickness of the connector can ensure superior seismic performance while saving resources. It follows from the parameter analysis of FEM2 and FEM3-8 that the initial stiffness and the bearing capacity of the specimens with a greater thickness and plate length were slightly higher. The best energy dissipation capacity was achieved when using a 6 mm thickness or a 50 mm plate length in different parameter groups. It is worth mentioning that this finite element analysis method can provide a theoretical guide for future research and implementation of such connectors.

## Figures and Tables

**Figure 1 materials-15-08778-f001:**
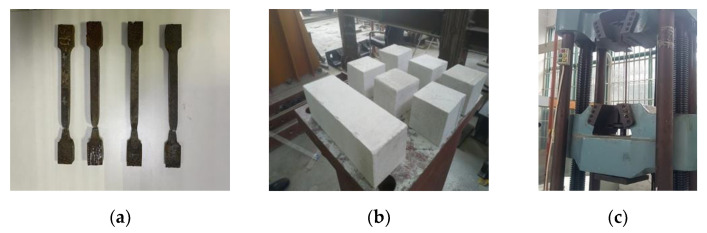
Details of the material properties test. (**a**) Tensile coupons; (**b**) ALC blocks; (**c**) universal hydraulic testing machine.

**Figure 2 materials-15-08778-f002:**
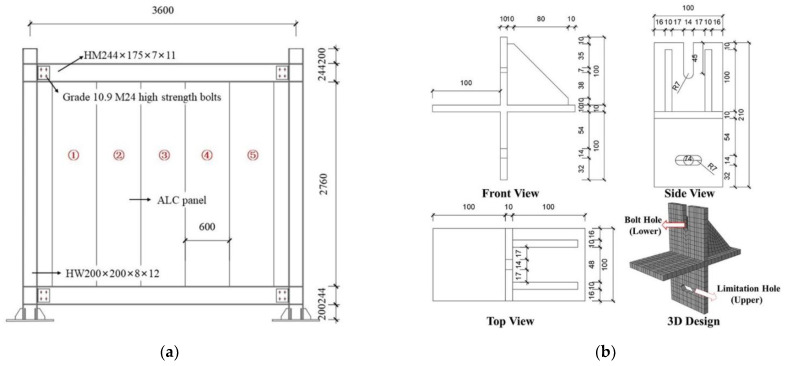
Details of the experimental design. (**a**) Geometric dimension of the flat combination frame; (**b**) geometric dimension of the slip-type crossing connector.

**Figure 3 materials-15-08778-f003:**
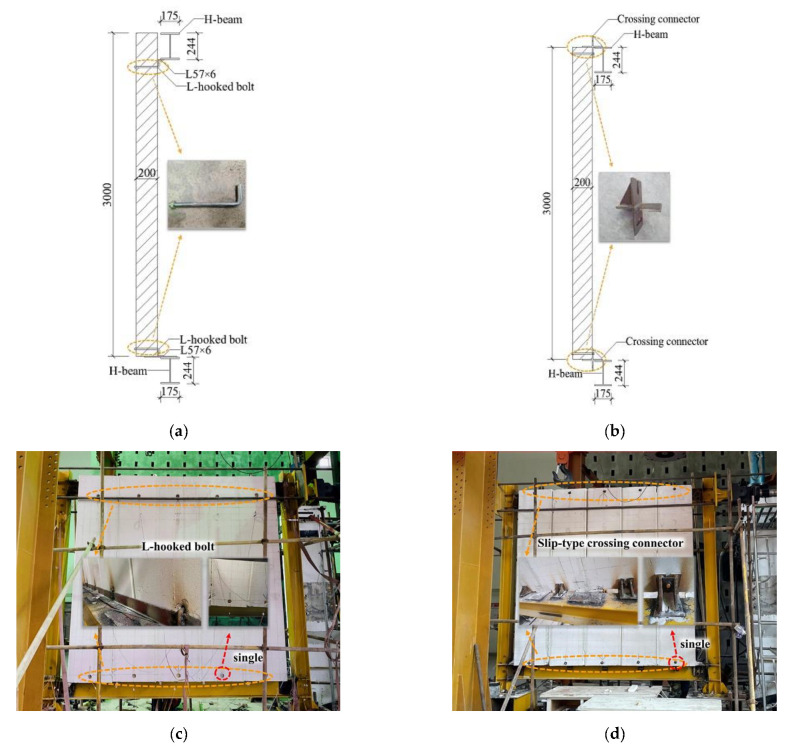
Connection types of walls to steel beams and the site of testing. (**a**) Connection schematic diagram of L-hooked bolt; (**b**) connection schematic diagram of crossing connector; (**c**) specimen JD-1; (**d**) specimen JD-2.

**Figure 4 materials-15-08778-f004:**
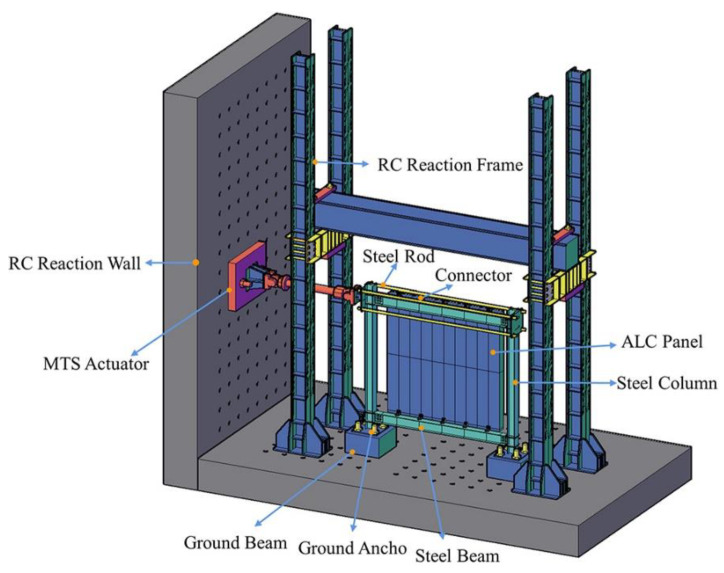
Loading device diagram.

**Figure 5 materials-15-08778-f005:**
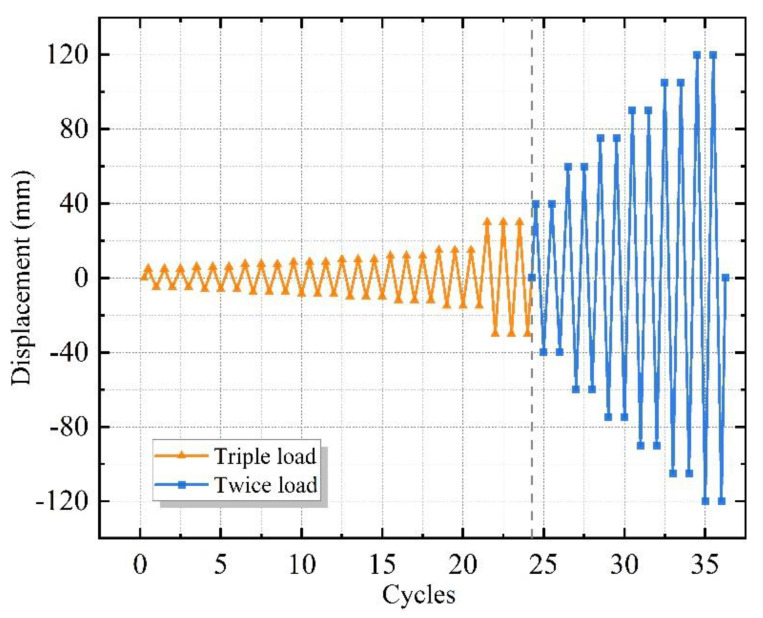
Test loading system.

**Figure 6 materials-15-08778-f006:**
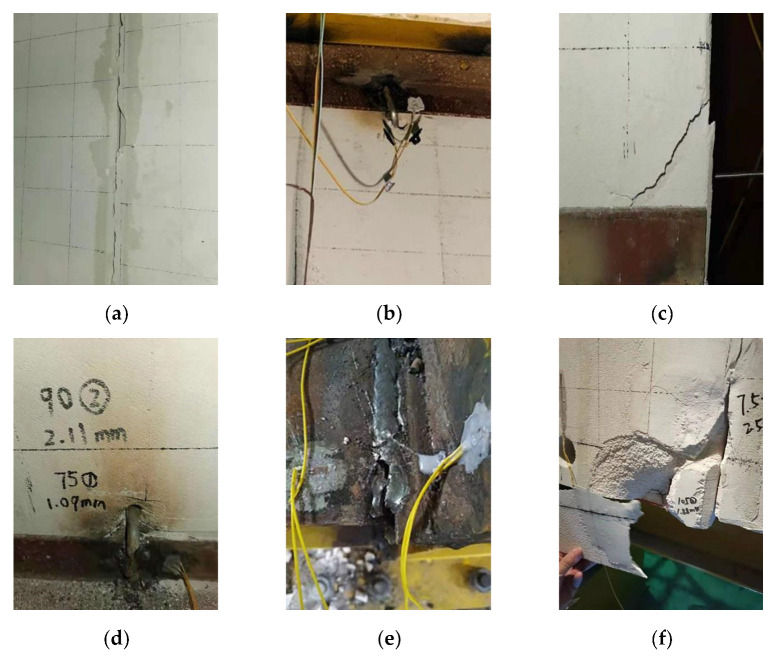
Phenomena during the test of JD-1. (**a**) Destruction of jointing mortar; (**b**) loosed L-hooked bolt; (**c**) crushing at the panel corner; (**d**) cracks at L-hooked bolt hole; (**e**) weld fracture; (**f**) spalling of ALC panels.

**Figure 7 materials-15-08778-f007:**
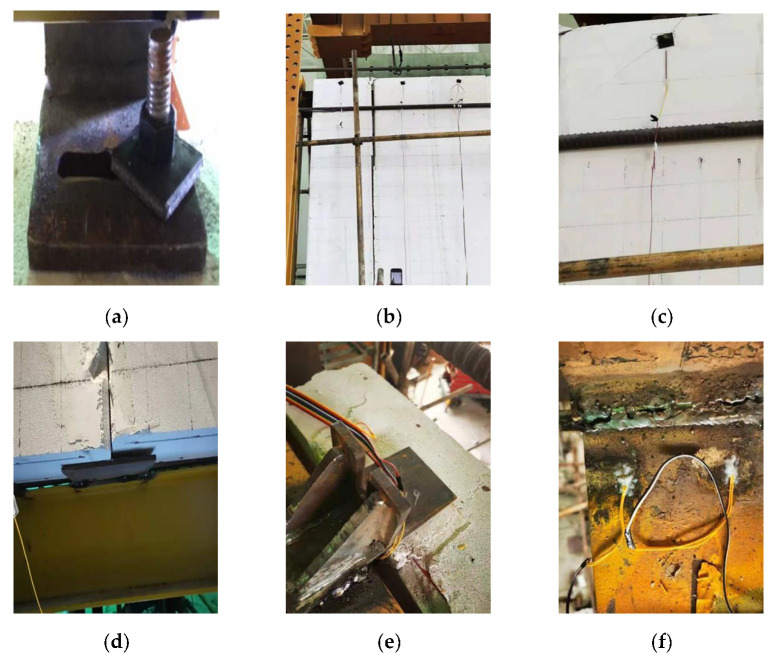
Phenomena during the test of JD-1. (**a**) Relative Sliding within bolt holes; (**b**) separating between the panels; (**c**) cracks at the bolt hole; (**d**) dislocation of ALC panels; (**e**) crushing of ALC panel near the upper connector; (**f**) weld fracture.

**Figure 8 materials-15-08778-f008:**
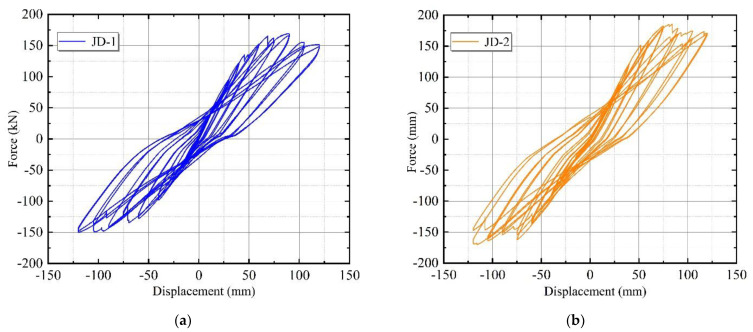
Load–displacement hysteric curves. (**a**) JD-1; (**b**) JD-2.

**Figure 9 materials-15-08778-f009:**
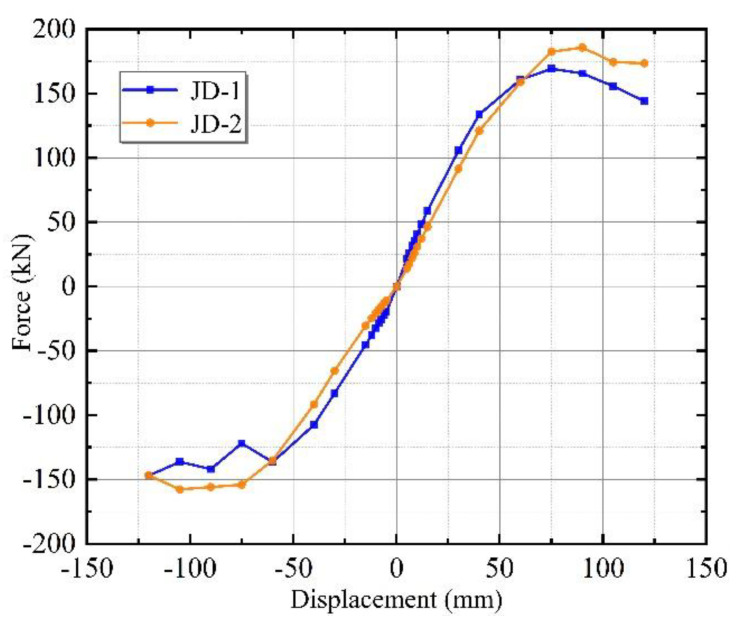
Skeleton curves of JD-1 and JD-2.

**Figure 10 materials-15-08778-f010:**
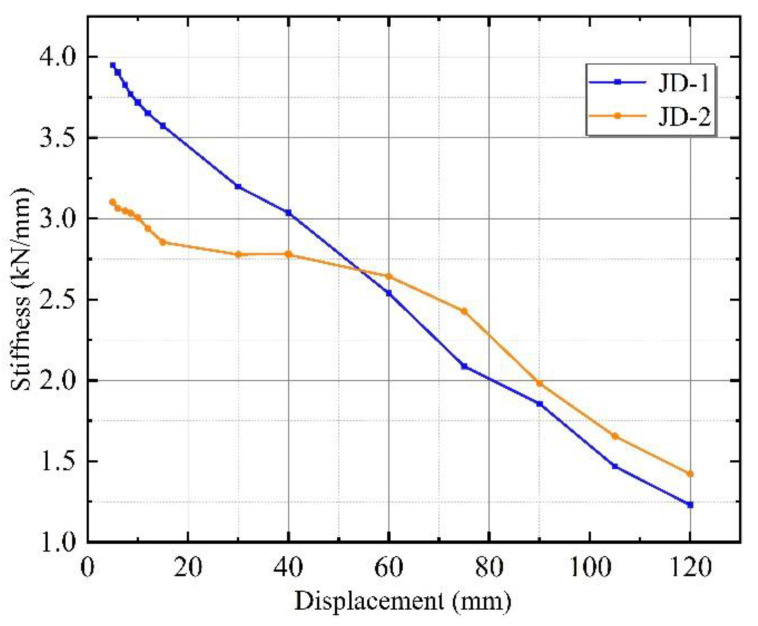
Stiffness degradation of JD-1 and JD-2.

**Figure 11 materials-15-08778-f011:**
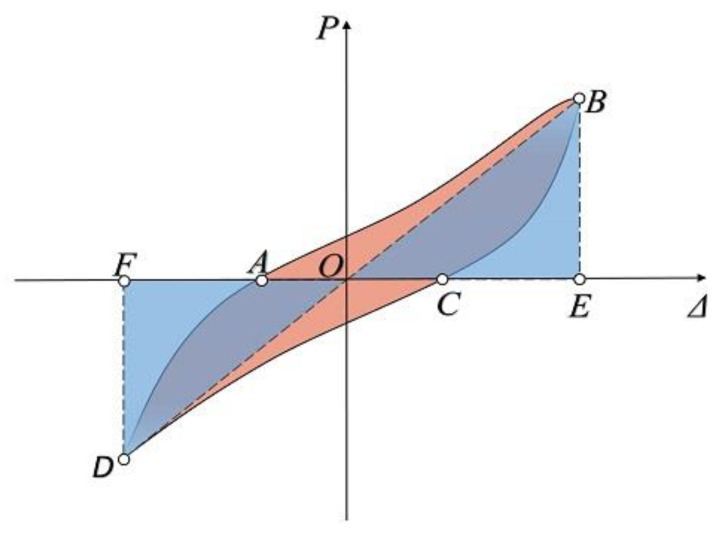
The dissipation capacity area.

**Figure 12 materials-15-08778-f012:**
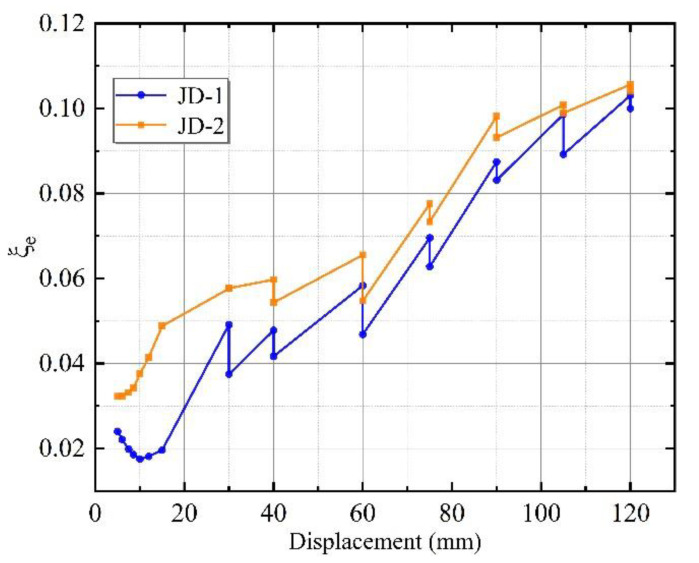
The equivalent viscous damp ratio of JD-1 and JD-2.

**Figure 13 materials-15-08778-f013:**
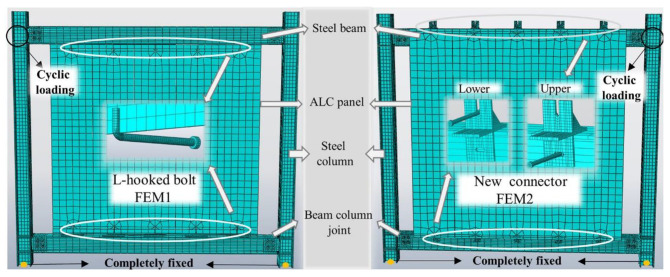
Finite element analysis model.

**Figure 14 materials-15-08778-f014:**
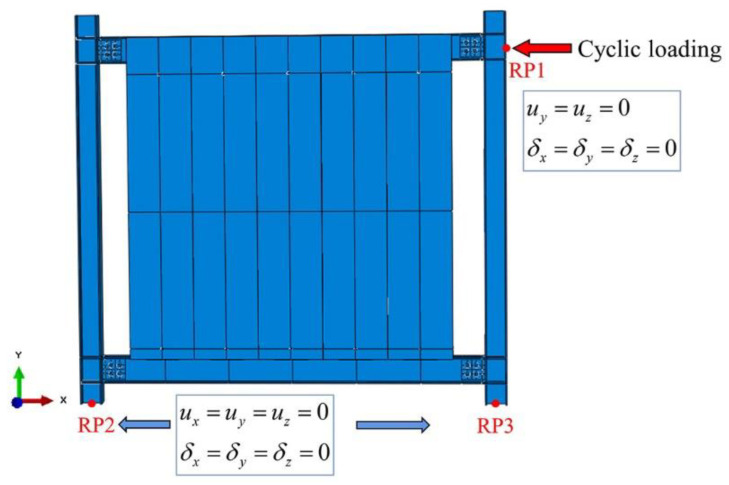
Loads and boundary conditions.

**Figure 15 materials-15-08778-f015:**
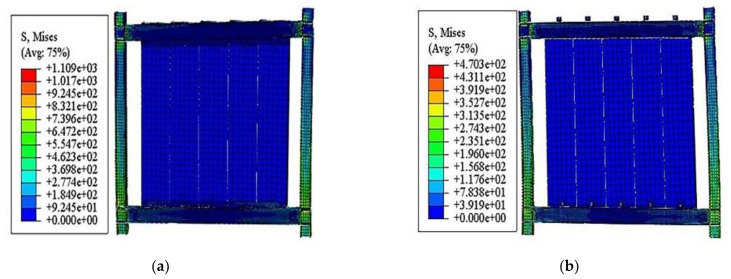
Comparing the stress cloud diagrams of the specimens and hysteresis curves of the test and simulation. (**a**) FEM1; (**b**) FEM2; (**c**) L-hooked bolt; (**d**) Slip-type crossing connector.

**Figure 16 materials-15-08778-f016:**
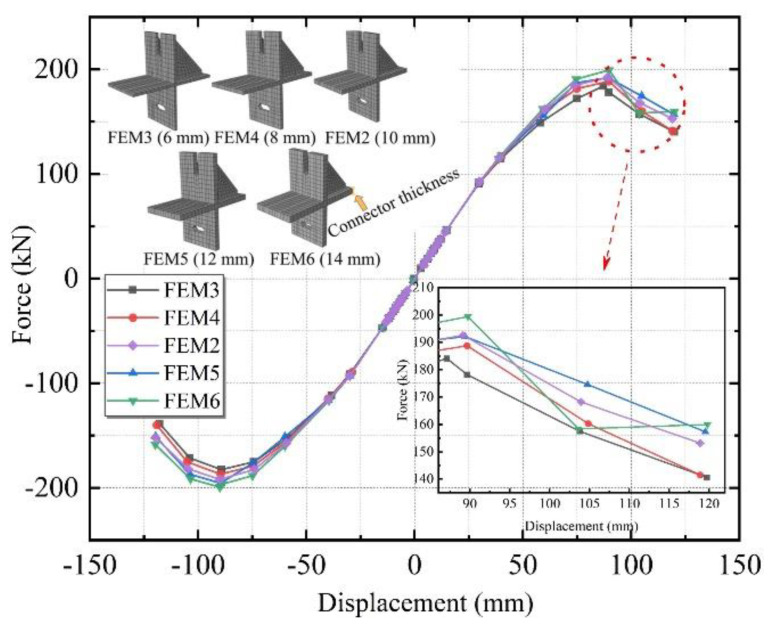
Skeleton curves with different specimen thicknesses.

**Figure 17 materials-15-08778-f017:**
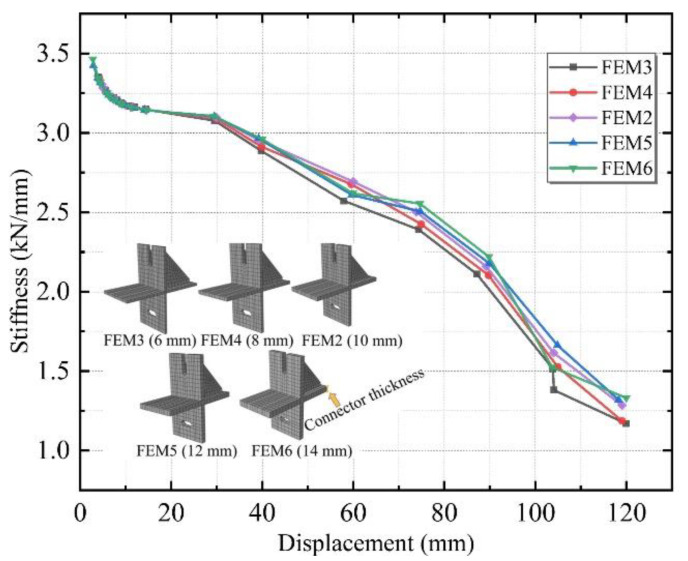
Stiffness degradation curves with different specimen thicknesses.

**Figure 18 materials-15-08778-f018:**
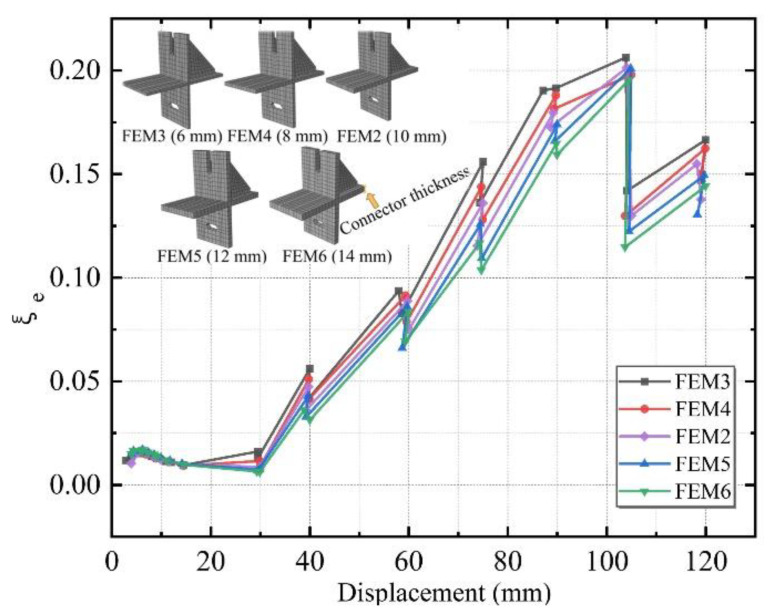
The equivalent viscous damp ratio for specimens with different thicknesses.

**Figure 19 materials-15-08778-f019:**
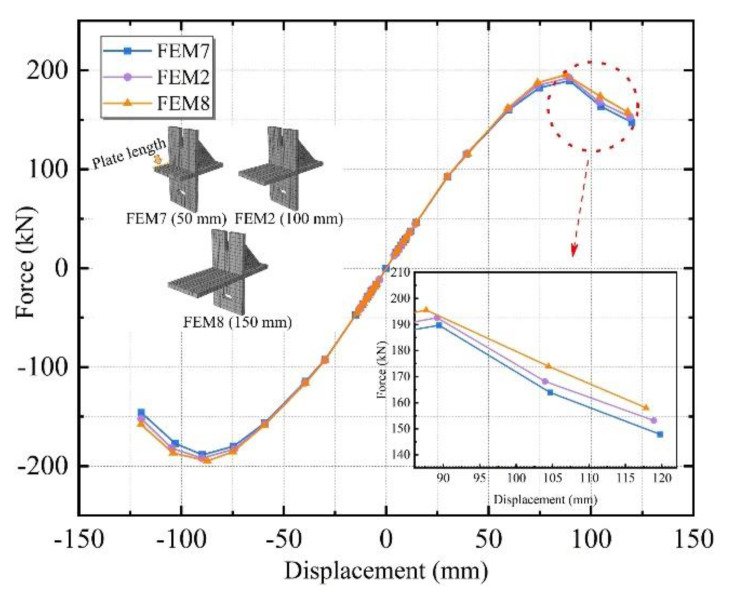
Skeleton curves of specimens with different plate lengths.

**Figure 20 materials-15-08778-f020:**
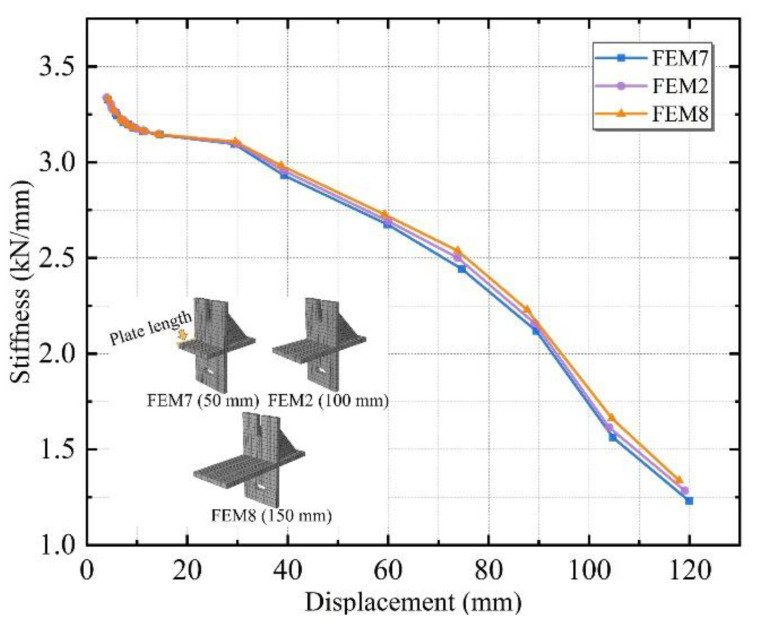
Stiffness degradation curves for specimens with different plate lengths.

**Figure 21 materials-15-08778-f021:**
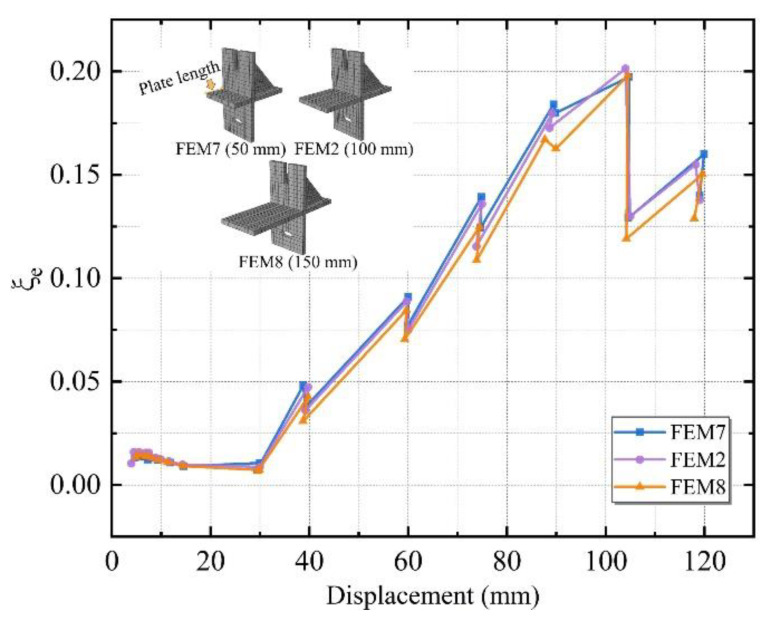
The equivalent viscous damp ratio for specimens with different plate lengths.

**Table 1 materials-15-08778-t001:** Material performance test results.

Specimen	Thickness (*t*/mm)	Yield Stress (*ƒ_y_*/MPa)	Ultimate Stress (*ƒ_u_*/MPa)	Elongation Stress (*δ*/%)
Connector	10	376.6	510.1	19.6
Beam web	7	275.3	411.3	22.3
Beam flange	11	263.4	401.6	25.2
Column web	8	278.2	409.8	20.8
Column flange	12	289.5	435.4	24.7

**Table 2 materials-15-08778-t002:** Parameters of the specimen.

Specimen	Wall Panel Type	Thickness/mm	Connection Method	Connector
JD-1	ALC panel	200	External mount	L-hooked bolt
JD-2	ALC panel	200	External mount	Slip-type crossing connector

**Table 3 materials-15-08778-t003:** Characteristic values on skeleton curves of specimens.

Specimen	Yield Point		Peak Point		Limit Point	
	Δ*_y_*/mm	*P_y_*/kN	Δ*_m_*/mm	*P_m_*/kN	Δ*_γ_*/mm	*P_γ_*/kN
JD-1	51.75	137.62	68.58	169.19	120	144.14
JD-2	58.77	156.59	84.17	185.68	120	173.48

**Table 4 materials-15-08778-t004:** Details of material properties.

Material Type		Density (*t*/mm^3^)	Modulus of Elasticity (MPa)	Poisson’s Ratio
Steel	Q235B	7.89 × 10^−9^	200,000	0.3
	Q345B	7.89 × 10^−9^	206,000	0.3
ALC		5 × 10^−10^	1770	0.2

## Data Availability

Not applicable.

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
