# Peer review of "Cyclic Behavior of Autoclaved Aerated Concrete External Panel with New Connector"

_materials, 2022, doi:10.3390/ma15248778_

Round 1
Reviewer 1 Report
This manuscript reviewed the ‘Cyclic behavior of autoclaved aerated concrete external panel with new connector’. The manuscript is elaborately described and contextualized with the help of previous and present theoretical background and empirical research. All the references cited are relevant to this area of research. The conclusions are supported by the results. However, some minor corrections need to be addressed before the acceptance the Manuscript.
1. Mention the need of this study in abstract.
2. Arrange keywords in alphabetical order.
3. How your study is different from existing studies?
4. Table 1. Use capital P for MPa.
5. Compare your results with existing literatures.
6. Why your study preferred ABAQUS rather than ANSYS?
7. State the recommendation for future research next to the conclusion.
8. More references are to be included.
Author Response
Response to Reviewer 1 Comments
Comments and Suggestions for Authors:
This manuscript reviewed the ‘Cyclic behavior of autoclaved aerated concrete external panel with new connector’. The manuscript is elaborately described and contextualized with the help of previous and present theoretical background and empirical research. All the references cited are relevant to this area of research. The conclusions are supported by the results. However, some minor corrections need to be addressed before the acceptance of the Manuscript.
Authors’ Response: We appreciate the acknowledgment of the reviewer for this paper. Several significant improvements in the revised manuscript have been made according to your valuable suggestions. Revised portions are marked in red in this paper.
Q1. Mention the need of this study in abstract.
Response: We sincerely thank you for your careful review. In our resubmitted manuscript, the need for this research has been mentioned in the abstract. We mentioned that the development of the new connector can improve the relevant deficiencies of existing connectors. A comparison of the pros and cons of existing connectors in the literature has been listed in this paper. And the parametric analysis model could help us improve the performance of the new connector and provide a reference for the development of such new connectors. (Page 1, Line 18-22).
Q2. Arrange keywords in alphabetical order.
Response: Thank you for your suggestion. As suggested by the reviewer, we have arranged keywords in alphabetical order. (Page 1, Line 23-24 with red color).
Q3. How your study is different from existing studies?
Response: Thank you for your careful review. We believe that the existing studies on the seismic performance of external panel connector often focuses on rigid connectors and has great limitations. We developed a flexible connector that can solve the existing deficiencies well. Different from other studies, we use finite element analysis software to conduct a parametric analysis of the new connector, and the research method can be an optimized solution for engineering applications.
Q4. Table 1. Use capital P for MPa.
Response: We are extremely grateful to the reviewer for pointing out this problem. We have corrected the problem and used capital P for MPa in Table 1. (Page 3, Line 91).
Q5. Compare your results with existing literatures.
Response: Thank you for your suggestion. As suggested by the reviewer, we compare the contributions of the connectors developed in the existing literature and point out their shortcomings. The comparison of the existing connectors with the results shows that the slip-type crossing connector is fully functional and reliable, and can solve construction problems. We have added more details in the Introduction Part. (Page 2, Line 57-77, Line 89-91).
Q6. Why your study preferred ABAQUS rather than ANSYS?
Response: Thank you for your careful review. In our research work, the seismic aspect is inseparable from all kinds of nonlinear problems, such as the contact behavior of components, material damage behavior under reciprocating action, overall buckling behavior of components, and large deformation behavior in the process of failure. When dealing with these problems, ABAQUS can get more realistic and accurate results, while ANSYS is more difficult to converge when dealing with various nonlinear problems.
Q7. State the recommendation for future research next to the conclusion.
Response: Thank you for your suggestion. As suggested by the reviewer, we have stated the recommendation for future research next to the conclusion. We added that this FEMs analysis method in this text can be used as a theoretical guide for future study and implementation of such new connectors. (Page 18, Line 467-469).
Q8. More references are to be included.
Response: Thank you very much for your suggestion, which is very helpful for us to improve the manuscript. We have added more related references, this paper has cited them in the text, and we hope it would be a good way to enrich the content of the article. (Reference [1] ,[8] ,[17] ,[27] ,[28] ,[29], [33]).
We would like to thank the referee again for taking the time to review our manuscript. We really appreciate your efforts in reviewing our manuscript during this unprecedented and challenging time. We wish good health to you, your family, and your community. Your careful review has helped to make our study clearer and more comprehensive.

Reviewer 2 Report
· The informal language is not suitable and should be improved extensively. The article needs major grammatical and syntax improvements. Use of English service center is recommended. Several sentences are not clear and understandable.
· Majority of the qualitative statements should be modified for quantified result comparisons.
· The introduction needs to be revised for higher quality language. The authors mentioned some works without stating about the contributions, pros and cons and the how the current work would address.
· The purpose of the article should be clarified in details, why and where this study could be beneficent, more in depth conclusion should be provided.
· The authors mentioned “prefabricated building systems have become a promising candidate to meet the requirements of green buildings with good seismic performance and repairability while saving more resources.” The following reference could be added for comprehensiveness of this statement: Analysis and design recommendations for corrugated steel plate shear walls with a reduced beam section.
· The authors mentioned “Many scholars concentrated on the seismic performance of various precast concrete elements and demonstrated their various mechanical qualities.” The following reference could be added for comprehensiveness of this statement: Compressive behavior of concrete under environmental effects. IntechOpen.
· For the Energy dissipation section, the authors mentioned “the seismic release energy well is an important indicator of its seismic performance”, related reference should be added.
· More descriptive legends and high quality figures are needed,
· Equation used previously should be clearly referenced.
· More in depth conclusions should be drawn based on various studies, the summary should indicate in depth results and conclusions.
· Figures needs to be professionally done and caption should be more descriptive. Fig 13 should have details of boundary condition, mesh and model specifications.
· Why the simulation analysis and the actual results from Figure 15 are different significantly, especially on the unloading regime.
·
Author Response
Response to Reviewer 2 Comments
Comments and Suggestions for Authors:
Q1. The informal language is not suitable and should be improved extensively. The article needs major grammatical and syntax improvements. Use of English service center is recommended. Several sentences are not clear and understandable.
Response: Thank you for your careful review. We are very sorry for the mistakes in this manuscript and the inconvenience they caused in your reading. The manuscript has been thoroughly revised and rewritten through the journal's recommended English service, so we hope it can meet the standard and the revisions in the text are shown using the red highlight for additions.
Q2. Majority of the qualitative statements should be modified for quantified result comparisons.
Response: Thank you for pointing out this problem in the manuscript. As suggested by the reviewer, we have modified the relevant qualitative statements in the test results analysis and parameter analysis section (Subsections 3.2 and 3.4). Especially in the Conclusion Part (Section 5), we rewrite the content of quantified result comparisons, and the readers would know more details about the depth conclusion.
Q3. The introduction needs to be revised for higher quality language. The authors mentioned some works without stating about the contributions, pros and cons and the how the current work would address.
Response: Thank you for your precious comments and advice. As suggested by the reviewer, we have compared the contributions, pros, and cons of mentioned works, which proves that the existing researches on the connectors have its limitation. In order to solve the relevant deficiencies of the existing connectors, we design a new flexible connector. More details have been added in the Introduction Part. (Page 2, Line 57-91 with red color)
Q4. The purpose of the article should be clarified in details, why and where this study could be beneficent, more in depth conclusion should be provided.
Response: Thanks very much for your comments. We believe that the connection behavior and structural measures of the wall and the main structure are related to the safety and stability of the whole structure. In the prefabricated high-rise steel structure building, the development of new connectors should ensure good deformation coordination ability and seismic performance. On this basis, the relevant research on the new connector is beneficial. We have added more details in the Introduction Part. (Page 2, Line 51-55, Line 77 with red color)
Q5. The authors mentioned “prefabricated building systems have become a promising candidate to meet the requirements of green buildings with good seismic performance and repairability while saving more resources.” The following reference could be added for comprehensiveness of this statement: Analysis and design recommendations for corrugated steel plate shear walls with a reduced beam section.
Response: Thank you very much for your suggestion, which is very helpful for us to improve the manuscript. We have read the article entitled “Analysis and design recommendations for corrugated steel plate shear walls with a reduced beam section” carefully. We are surprised to read this article and thought it was related to the topic of our article, this paper has cited it in the Introduction Part.
Q6. The authors mentioned “Many scholars concentrated on the seismic performance of various precast concrete elements and demonstrated their various mechanical qualities.” The following reference could be added for comprehensiveness of this statement: Compressive behavior of concrete under environmental effects. IntechOpen.
Response: Thank you for your careful review. The important piece of research found the compressive behavior of concrete under environmental effects. We cited the article in the Introduction Part.
Q7. For the Energy dissipation section, the authors mentioned “the seismic release energy well is an important indicator of its seismic performance”, related reference should be added.
Response: Thank you for the suggestion. We have added the related reference, the related research found the new-type composite exterior wallboard absorbs energy when loading and releases energy when unloading and the seismic performance of specimens is mainly dependent on the capacity of energy dissipation. We cited the article in the text. (Page 10, Line 259)
Q8. More descriptive legends and high quality figures are needed.
Response: Thank you for pointing out this problem in our manuscript. According to the reviewer’s comment, we have revised the figures and graphs to show the information in the charts clearly. We have adjusted the charts to ensure readers get more content from the description.
Q9. Equation used previously should be clearly referenced.
Response: Thank you for your precious comments and advice. We have added the information required as explained, the equation used previously has been referenced clearly. (Page 9, Line 233; Page 10, Line 262)
Q10. More in depth conclusions should be drawn based on various studies, the summary should indicate in depth results and conclusions.
Response: We appreciate the reviewer’s suggestion deeply. According to the reviewer’s comment, we make significant improvements in the Conclusion Part and show the results and conclusions in more depth. More details have been shown in the Conclusion Part. (Page 17-18, Line 438-469 with red color)
Q11. Figures needs to be professionally done and caption should be more descriptive. Fig 13 should have details of boundary condition, mesh and model specifications.
Response: We gratefully appreciate your valuable suggestion. According to the revised content, we have modified figures to clearly show the information in the chart. Especially in the test results part, we added more details of captions to ensure that the reader gets more information from the description. According to the reviewer’s comment, we have redrawn Figure 13 to include the above contents in more detail.
Q12. Why the simulation analysis and the actual results from Figure 15 are different significantly, especially on the unloading regime.
Response: Thanks very much for your comments. The main reason for this is that the finite element simulation environment is more ideal than the test environment. The complex environment such as the slippage of the floor beams and the gaps between the components in the test was simplified in the FEMs simulation. Ignoring the influence of extraneous factors, the FEMs software simulation results are in good agreement with the test results. We try to refine the results in the future by researching different approaches or different material laws.
We would like to thank the referee again for taking the time to review our manuscript. We really appreciate your efforts in reviewing our manuscript during this unprecedented and challenging time. We wish good health to you, your family, and your community. Your careful review has helped to make our study clearer and more comprehensive.

Reviewer 3 Report
The present research study a relevant topic, the connections between prefabricated elements and their performance under cyclic loadings. The authors have presented a study, combining experimental and advanced numerical simulations.
- The numerical models should be better explained, especially related to relevant nonlinear parameters.
- The figures and graphs should be modified and enhanced their presentability. all the letters/numbers should be readable. It is recommended to arrange the data of the graphs based on a progressive increase of the plate thickness.
- Some minor grammar errors are present and should be removed.
- The literature review of the article is very poor. The authors should improve it by adding other relevant research articles.
- The conclusions are not well organized. Generally speaking, they resemble a short description of the results. I would suggest rewrite this section. It would be more relevant to emphasize the objective of the research and what the authors have concluded by their research. It is mandatory to avoid trivial information.
Author Response
Response to Reviewer 3 Comments
Comments and Suggestions for Authors:
The present research study a relevant topic, the connections between prefabricated elements and their performance under cyclic loadings. The authors have presented a study, combining experimental and advanced numerical simulations.
Response:
Thank you very much for reviewing this article. We have carried out detailed proofreading of the manuscript. Several significant improvements in the revised manuscript have been made according to your valuable suggestions. Revised portions are marked in red in this paper.
Q1. The numerical models should be better explained, especially related to relevant nonlinear parameters.
Response: Thanks for your comments. Considering the reviewer’s suggestion, we have explained the finite element model in detail. In our research work, the seismic aspect must be inseparable from all kinds of nonlinear problems, such as the contact behavior of components, and material damage behavior under reciprocating action. We have added more details about friction behavior and the plastic damage model in the article, and it would be helpful to explain the numerical models. (Page 11, Line 301-305, Page 13, Line 315-318 with red color)
Q2. The figures and graphs should be modified and enhanced their presentability. all the letters/numbers should be readable. It is recommended to arrange the data of the graphs based on a progressive increase of the plate thickness.
Response: Thank you for pointing out this problem in our manuscript. According to the revised content, we have revised the figures and graphs to show the information in the charts clearly. We have adjusted the size of the letters and numbers for the reader's convenience. According to the reviewer’s comment, we have arranged the data of the graphs based on a progressive increase in the plate thickness.
Q3. Some minor grammar errors are present and should be removed.
Response: We apologize for the language problems in the original manuscript. The language presentation was improved and the manuscript has been thoroughly revised through the English service. Minor grammar errors have been removed and the revisions in the text are shown using the red highlight for additions.
Q4. The literature review of the article is very poor. The authors should improve it by adding other relevant research articles.
Response: Thank you for your precious comments and advice. According to the reviewer’s comment, we have added more related references in the literature review of the article. We have added reference [1] to the presentation of promising aspects of prefabricated building systems, and reference [8] found the compressive behavior of concrete under environmental effects. In order to better explain the wall and the main structure of the connection performance related to the safety and stability of the whole structure, we added a reference [17]. We thought they were related to the topic of our article, the paper has cited them in the Introduction Part. (Reference [1],[8],[17]).
Q5. The conclusions are not well organized. Generally speaking, they resemble a short description of the results. I would suggest rewrite this section. It would be more relevant to emphasize the objective of the research and what the authors have concluded by their research. It is mandatory to avoid trivial information.
Response: Thank you very much for your suggestion, which is very helpful for us to improve the manuscript. According to the reviewer’s comment, we rewrite the Conclusion Part and show the results and conclusions in more depth. In order to avoid trivial information, we emphasized the purpose of the research and refined the relevant conclusions. We have shown more details in the Conclusion Part. (Page 17-18, Line 438-469 with red color)
We would like to thank the referee again for taking the time to review our manuscript. We really appreciate your efforts in reviewing our manuscript during this unprecedented and challenging time. We wish good health to you, your family, and your community. Your careful review has helped to make our study clearer and more comprehensive.

Round 2
Reviewer 2 Report
English should be improved
Reviewer 3 Report
The authors have addressed correctly all the raised comments.
In my opinion the quality of the paper is significantly enhanced. All the aspects of the research, detailing of the experiments/numerical models, results and the conclusions are clearly presented.
I suggest publishing it.